# Characteristics and Risk Assessment of Environmentally Persistent Free Radicals (EPFRs) of PM_2.5_ in Lahore, Pakistan

**DOI:** 10.3390/ijerph20032384

**Published:** 2023-01-29

**Authors:** Mushtaq Ahmad, Jing Chen, Qing Yu, Muhammad Tariq Khan, Syed Weqas Ali, Asim Nawab, Worradorn Phairuang, Sirima Panyametheekul

**Affiliations:** 1State Key Joint Laboratory of Environment Simulation and Pollution Control, School of Environment, Beijing Normal University, Beijing 100875, China; 2Department of Environmental Engineering, Faculty of Engineering, Chulalongkorn University, Bangkok 10330, Thailand; 3Department of Science and Environmental Studies, The Education University of Hong Kong, Taipo, New Territories, Hong Kong, China; 4Department of Environmental Sciences, Abdul Wali Khan University, Mardan 23200, Pakistan; 5Faculty of Geosciences and Civil Engineering, Institute of Science and Engineering, Kanazawa University, Kanazawa 920-1192, Ishikawa, Japan; 6Thailand Network Centre on Air Quality Management: TAQM, Chulalongkorn University, Bangkok 10330, Thailand; 7Research Unit: HAUS IAQ, Chulalongkorn University, Bangkok 10330, Thailand

**Keywords:** EPFRs, carbonaceous species, hydroxyl radicals, risk assessment, oxidative potential

## Abstract

Environmentally persistent free radicals (EPFRs) are an emerging pollutant and source of oxidative stress. Samples of PM_2.5_ were collected at the urban sites of Lahore in both winter and summertime of 2019. The chemical composition of PM_2.5_, EPRF concentration, OH radical generation, and risk assessment of EPFRs in PM_2.5_ were evaluated. The average concentration of PM_2.5_ in wintertime and summertime in Lahore is 15 and 4.6 times higher than the national environmental quality standards (NEQS) of Pakistan and WHO. The dominant components of PM_2.5_ are carbonaceous species. The concentration of EPFRs and reactive oxygen species (ROS), such as OH radicals, is higher in the winter than in the summertime. The secondary inorganic ions do not contribute to the generation of OH radicals, although the contribution of SO_4_^2+^, NO_3_^−^, and NH_4_^+^ to the mass concentration of PM_2.5_ is greater in summertime. The atmospheric EPFRs are used to evaluate the exposure risk. The EPFRs in PM_2.5_ and cigarette smoke have shown similar toxicity to humans. In winter and summer, the residents of Lahore inhaled the amount of EPFRs equivalent to 4.0 and 0.6 cigarettes per person per day, respectively. Compared to Joaquin County, USA, the residents of Lahore are 1.8 to 14.5 times more exposed to EPFRs in summer and wintertime. The correlation analysis of atmospheric EPFRs (spin/m^3^) and carbonaceous species of PM_2.5_ indicates that coal combustion, biomass burning, and vehicle emissions are the possible sources of EPFRs in the winter and summertime. In both winter and summertime, metallic and carbonaceous species correlated well with OH radical generation, suggesting that vehicular emissions, coal combustion, and industrial emissions contributed to the OH radical generation. The study’s findings provide valuable information and data for evaluating the potential health effects of EPFRs in South Asia and implementing effective air pollution control strategies.

## 1. Introduction

Air pollution is the world’s leading environmental concern and threatens human health [1] and the natural ecosystem [2]. Indeed, 3.2 million people died worldwide due to air pollution, the number of PM_2.5_-related deaths in China increased by 0.39 million (23%) in the 15 years between 2002 and 2017 [3]. The fine particles (PM_2.5_) act as carriers for pathogens, free radicals, and heavy metals. Acute and chronic exposure to PM air pollution is associated with increased risk of death from cardiovascular diseases, including ischemic heart disease and heart failure [4]. The chemical composition of particulate matter can significantly influence these responses [5]. The chemical composition of PM may differ significantly depending on emission sources, weather, and the possibility of dispersion [6]. The well-being of the natural ecosystem and human health have been adversely affected by gaseous pollutants, e.g., CO, SO_2_, NO_3_, and O_3_ [7]. Human health is primarily affected by combustion-derived PM, ultrafine particles, and metal- and PAH-enriched particles with high oxidative potential. Secondary inorganic particles, such as ammonium, sulfate, and nitrate, have significantly weaker evidence of adverse health effects [8].

Environmentally persistent free radicals (EPFRs) in atmospheric particles have drawn significant attention as emerging pollutants in recent years. EPFRs are among the most significant sources of oxidative stress [9]. Atmospheric particulates can contain various EPFRs types. In the atmosphere, the EPFRs produces in the particles have been identified by studies as phenoxyl, semiquinone, and other EPFRs types [10]. The EPFRs types can be distinguished by the g factor parameter [11]. The g factor value of oxygen-centered radicals is higher than 2.004, carbon-centered free radicals is less than 2.003, while the carbon-centered radicals together with oxygen atoms range from 2.003 to 2.004 [12]. The g factor value of EPFR range from 2.0030 to 2.0047 in atmospheric particles depending on PM sources and its chemical composition [13]. The presence of oxidative stress-causing species in particulate matter in human-influenced regions has been reported in several studies. These species are correlated with organic compounds and metals, including quinones, Cr, and Zn [14]. Recent studies have highlighted the adverse effects of EPFR on the environment and human health [15]. The health effects induced by exposure to PM and the combustion by-products that are adsorbed on them mainly manifest in elderly people with pre-existing cardiovascular diseases [16], neurodegenerative diseases [17], increased irritation of the eye, and respiratory infections [18]. The continuous conversion of O_2_ molecules into reactive oxygen species (ROS) by EPFR is a possible mechanism for such health effects [13]. PM_2.5_ contains EPFRs emitted into the atmosphere by thermal processes that induce the formation of harmful reactive oxygen species (ROS). These species can adversely affect human health and can cause DNA damage [19]. The OH radicals are a highly reactive species, able to oxidize almost any organic compound rapidly and effectively. The generation of free radicals serves as the controlling mechanism for chemical reactions [20].

In Europe, North America, and Asia, particularly rapid economic development countries, such as China and India, have conducted the most detailed research studies on airborne toxic heavy metals than any other region [21,22,23,24]. The studies conducted in the metropolitan regions of China focused primarily on the sources and major components of PM_2.5_ [25] and heavy metal concentration and health risk assessment [26]. Soil dust, volcanic eruptions, forest fires, and meteoric dust are the natural sources of metal elements in the atmosphere. Fuel oil burning, coal combustion, and other metallurgical processes are the major anthropogenic sources of heavy metals [27]. Due to the low density, larger surface area per unit volume, and high organic matter content of PM_2.5_, most of the identified elements and metals have been reported in PM_2.5_ [28]. PM_2.5_-bound heavy metals can easily be re-suspended and persist for longer periods of time [29]. In addition, these heavy metals can easily enter the body via ingestion, inhalation, and dermal contact, adversely affecting human health [30].

This study aims to determine the chemical composition of PM_2.5_, its oxidative potential, and the EPFR concentration in PM_2.5_. The risk assessment of atmospheric EPFRs was determined by converting the amount of EPFR inhaled to the number of cigarettes smoked per person per day. The correlation analysis was duly carried out to identify the possible sources of OH radicals and EPFRs. This research work identified the concentration of EPFRs and ROS in particulate matter (PM), the oxidative stress caused by ROS, and their adverse effects on human health. This research will answer (1) What are the major components of PM_2.5_ and the concentration of PM_2.5_ in both wintertime and summertime, in the urban sites of Lahore, Pakistan? (2) What are the probable sources of the EPFRs and OH radicals and their concentration in wintertime and summertime? (3) How can the exposure risk of EPFRs be evaluated in both sampling periods in Lahore, Pakistan? This research work is the first in Pakistan to report EPFRs in PM_2.5_ and its exposure evaluation. The findings of this study provide the primary data for further evaluation of EPFRs in PM_2.5_ and their potential health effects in other mega-cities of Pakistan.

## 2. Methods and Materials

### 2.1. Site and Sampling Information

Diurnal samples of PM_2.5_ were collected at the urban sites of Lahore, Pakistan. On pretreated quartz fiber filters (PallFlex, Putnam, CT, USA, 90 mm), the samples of ambient PM_2.5_ were collected using an air volume sampler (HY-100D, Qingdao Hengyuan, Qingdao, China) with a flow rate of 30–100 L/min. The quartz fiber filters were baked in a furnace to burn the organics at 800 °C. Each sample was collected by drawing around 52 to 62 m^3^ of ambient air for 11:30 h each for daytime and nighttime. For filter weight, each filter was weighted twice before and after sampling. Before analysis, the samples were stored in a refrigerator at −20 °C. At each sample site, the air volume sampler was installed on a building about 5–10 m above the ground.

### 2.2. EPFRs Analysis

Environmentally persistent free radicals can be determined using electron paramagnetic resonance (EPR). The EPR method has the potential to determine radical types (g factor) and radical concentration. The filter samples were cut into three pieces (5 × 28 mm) to directly evaluate EPFRs using an EPR spectrometer. For EPFRs analysis, the parameters for measuring EPR were set as: microwave frequency of 9.1 GHz, power of 0.9 mW, the scan time of 240 s, and modulation amplitude of 0.2 mT, a quantitative analysis was carried out by using the software WINEPR. The total sample volume was used to estimate the atmospheric concentration of EPFRs (spins/m^3^). In contrast, the EPFR concentration in PM_2.5_ mass (spins/g) was determined by dividing the total number of spins by the mass of PM_2.5_ [31].

### 2.3. Oxidative Potential Analysis

A quarter of each filter was ultrasonically extracted in 10 mL of ultra-pure water for 20 min. Insoluble particles were removed by filtering each aqueous extract through a Teflon (PTFE) filter with a pore size of 0.45 μm using a syringe. A standard solution for OH radical generation was prepared by mixing pure water with 0.2, 0.4, 0.6, 0.8, and 1 mL of diluted hydroxyterephthalic acid (OHTA) solution. A mixture of 2 mL of diluted dithiothreitol (DTT) solution, 4 mL of sample extract, and 2 mL of disodium terephthalate (TPT) solution was incubated at 37 °C using a thermo-mixer. A 0.5 mL of dimethyl sulfoxide (DMSO) solution was mixed with 1.5 mL of the reaction solution at the regular intervals of 0, 12, 24, 36, 48, and 60 min to quench the OH radical generation. TPT captures OH radicals and generates a fluorescent compound, i.e., 2-hydroxyterephthalic acid (2-OHTA). The 2-OHTA fluorescence intensity was measured at an excitation/emission wavelength of 310/425 nm using a Fluorescence Spectrophotometer (Horiba Scientific; Edison, NJ, USA). The 2-OHTA concentration was determined by calibrating the instruments with a known concentration of a standard. The linear regression equation with (R^2^ = 0.98) was used to estimate the rate of OH radical generation. OH radical generation in the DTT assay was measured separately, following [32].

### 2.4. Other Analysis

The carbonaceous species, water-soluble ions, and metals in the PM_2.5_ samples were analyzed. Carbonaceous species that were measured included organic carbon (OC), elemental carbon (EC), and water-soluble organic carbon (WSOC). The detailed measurements of OC, EC, WSOC, and ions are reported in our previous study [33]. A Desert Research Institute (DRI) thermal/optical carbon analyzer was used to evaluate OC and EC, following the thermal/optical transmittance (TOT) protocol. The WSOC concentration was assessed using a TOC analyzer (TOC-L Shimadzu).

Li, Al, V, Cr, Mn, Mg, Fe, Cu, Zn, Ga, Cd, Pb, Be, Ti, As, Bi, Sn, Sr, Ba, Tl, Co, and Ni were among the measured metals. The detail of the dilute acid solution preparation is reported in our previous study [34]. Inductively coupled plasma-mass spectrometry (iCAP TQ ICP-MS, Thermo Scientific, Waltham, Massachusetts, United States) was then used to measure the metals in the solution.

### 2.5. Atmospheric EPFR Exposure Evaluation

The number of cigarettes smoked per person per day was used to determine the potential health risk of EPFR exposure. The equation used to convert the concentration of EPFR in PM_2.5_ to the number of cigarettes per person per day [35] is as follows:(1)$$N_{\mathrm{cig}}=\frac{\left(A_{\mathrm{EPFRs}}\cdot \;V\right)}{\left(R_{\mathrm{cig}}\cdot C_{\mathrm{tar}}\right)}$$
where the number of cigarettes per person per day is denoted by N_cig_, atmospheric EPFR concentration in PM_2.5_ is represented by A_EPFRs_ (spins/m^3^), and V is the amount of air per day inhaled by an adult, which is (20 m^3^/day) [36]. R_cig_ represents the free radical concentration in cigarette tar, which is (4.75 × 10^16^ spins/g) [37], and the amount of tar in a cigarette is represented by C_tar_, which is (0.013 g/cig) [35].

### 2.6. Statistical Analysis

The correlation between the components of PM_2.5_ and EPFRs and OH radicals was evaluated using a statistical package for the social sciences (SPSS) V 20.0. Differences were considered significant at *p* < 0.05. The correlation analysis between EPFRs and the concentration of the carbonaceous species (TC, OC, EC, WSOC, SOC) was carried out to identify the possible sources of EPFR. In contrast, the carbonaceous species and elements were used to carry out the correlation of OH radicals to determine their potential sources.

## 3. Results and Discussions

### 3.1. PM_2.5_ Mass and EPFRs Concentrations

In Table 1, the mean concentration of atmospheric EPFR and carbonaceous species of PM_2.5_ at Lahore is shown. The mean concentrations of PM_2.5_ at Lahore in wintertime and summertime sampling were 522.2 ± 222.0 μg/m^3^ and 162.5 ± 50.6 μg/m^3^, which were approximately 3.2 times higher in the winter than in the summer. Weather conditions and local anthropogenic activities increased the PM_2.5_ concentration in both sampling periods. Due to poor meteorological conditions in the wintertime at Lahore, the PM_2.5_ concentration is several times higher than in the summertime [38]. The dispersion of PM in wintertime is reduced due to the low inversion layer, wind speed, temperature, and humidity. As a result, pollutants accumulate near the surface, and the concentration of PM_2.5_ increases [39]. The high concentration of PM_2.5_ in the wintertime can be attributed to vehicular emissions, industrial activities, biomass burning, and increased heating activity [40]. The national environmental quality standards (NEQS) for ambient PM_2.5_ in Pakistan are 35 μg/m^3^ [41,42]. In winter and summertime, the average concentration of PM_2.5_ at Lahore was 15 and 4.6 times higher than the NEQS of Pakistan. In Beijing, Xi’an, and Xuanwei, the mass concentration of PM_2.5_ was in the range of 21–365 μg/m^3^ [43], 22–179 μg/m^3^ [44], and 21–50 μg/m^3^ [45], all of which were significantly lower than the concentration found in the current study. The mean concentration of PM_2.5_ in various seasons in northern Africa, such as in Kenitra city, Morocco 50.7 μg/m^3^ [46], and Bou-Ismail, in Tipaza, Algiers 6.94 μg/m^3^ (January), and 20.19 μg/m^3^ (July) [47], were reported as significantly lower than the current study. As shown in Table 2, the PM_2.5_ mass concentration in the current study was relatively higher than the previous studies conducted in urban areas of Pakistan and India [33,48,49,50,51,52,53].

As shown in Table 1, the mean concentration of atmospheric EPFR in both winter and summertime was 1.2 × 10^14^ spin/m^3^ and 1.7 × 10^13^ spin/m^3^, respectively. Comparatively, the concentration of atmospheric EPFRs at Lahore during the wintertime were significantly higher than in the summertime. As shown in Figure 1, the concentrations of EPFRs at Lahore varied from 2.9×10^13^−2.9×10^14^ spin/m^3^ and 2.9×10^12^−4.6× 10^13^ spin/m^3^ during the winter and summertime, respectively. These variations are attributed to industrial activities, vehicular emissions, and weather conditions in wintertime. The EPFR concentration in PM_2.5−1_ at Beijing was reported to be in the range of 1.0×10^15^−1.4×10^16^ spin/m^3^, and in PM_1_, the EPFR concentration ranged from 2.5× 10^15^ to 3.5×10^16^ spin/m^3^ [43]. In the current study, EPFRs were measured in particles, whereas at Beijing, EPFRs were measured in aqueous extract, and EPFR concentrations in the current study were lower than the Beijing. In contrast, the atmospheric concentrations of EPFRs in the PM_2.5_ at Erenhot, Zhangbei and Jinan, China, 1.6 × 10^13^, 5.7 × 10^13^, and 4.6 × 10^13^ spin/m^3^, respectively [54], were slightly lower than Lahore. The EPFR concentration reported in Wanzhou, Chongqing, China was 7.0 × 10^13^ spin/m^3^ [55]. The concentration of EPFRs in Nanjing was also slightly lower than in the current study. The concentration of EPFRs ranged from 2.78 × 10^12^ to 1.72 × 10^13^ spins/m^3^, with an average value of 7.61 × 10^13^ spins/m^3^ [56].

As shown in Table 1, the mean EPFR concentration in PM_2.5_ in winter and summertime in Lahore was 2.3×10^17^ spin/g, and 1.1×10^17^ spin/g. Several brick kilns and industrial activities frequently burn coal in the surroundings of Lahore [57]. As a result, a significant amount of EPFRs is emitted by coal combustion, and thus the higher concentration of EPFRs is reported at Lahore during wintertime [9]. At Xi’an, the EPFR concentration in PM_2.5_ during the winter season was in the range of 2.5×10^17^−8.1×10^18^ spin/g with an average concentration of 2.1×10^18^ spin/g [44]. The concentration of EPFRs at Xi’an was significantly higher than in the current study. The EPFR concentration in PM_2.5_ at Baton Rouge, Louisiana, USA, was reported in the range of 2.5×10^16^−2.8× 10^17^ spin/g [35], which is lower than in the current study. Another study was conducted at Erenhot, Zhangbei, and Jinan in China, and the EPFR concentration in PM_2.5_ reported was 3.7 × 10^17^, 1.1×10^18^, and 4.6×10^17^ spin/g [54]. In contrast, the EPFR concentrations in PM_2.5_ in current study were significantly lower than those reported in Chinese cities. The *g*-value of a radical is a dimensionless number that is indicative of the types of radical present. The *g*-value measures how the magnetic environment of the unpaired electrons differs from that of a free, gas-phase electron (*g* = 2.0023). The “*g*-value” is denoted as the center of an EPR spectrum in dimensionless unit [10]. For organic radicals, the *g*-values are typically quite near that of a free electron, ranging from 1.99 to 2.01. For instance, the *g*-value of the stable organic radical is 2.0036. In both sampling periods, the EPR spectra of EPFRs were compared, and the g factor was similar, suggesting that the EPFR type at Lahore in winter and summertime is the same. The EPFRs of tar and smoke from tobacco, coal, and petroleum have a g factor of ~2.0032 [58]. At Lahore, the g factor of EPFR in the wintertime varied from 2.0027 to 2.0032, while in the summertime, it varied from 2.0026 to 2.0033, with an average of 2.0030 and 2.0029 for winter and summer, respectively. The average g factor for Erenhot, Zhangbei, and Jinan, China, was (2.0031), similar to the current study [54]. In both sampling periods, the g factor of EPFRs includes carbon-centered EPFRs containing heteroatoms [31]. The average ∆H_p−p_ of EPFRs in summer and wintertime at Lahore was 6.2 and 4.6, which is significantly different. The ∆H_p−p_ value at Erenhot, Zhangbei, and Jinan, China was 3.8, 4.3, and 3.7, respectively [54]. The ∆H_p−p_ value at Zhangbei was in consistent with the ∆H_p−p_ value of summertime sampling of Lahore. In contrast, the ∆H_p−p_ value of EPFR in PM_2.5_ from Xian was in the range of 4.98−5.29 [31], which is slightly lower than the current study. In both sampling periods, the difference in ∆H_p−p_ indicated different sources of atmospheric EPFRs. The g factor and ∆H_p−p_ of EPFRs in both sampling periods at Lahore revealed that EPFRs are carbon-centered. In contrast, their chemical composition can be different due to different emission sources.

### 3.2. Carbonaceous Species

Table 1 shows the average concentrations of PM_2.5_ and its carbonaceous species in Lahore during the winter and summertime sampling. The variation of OC and EC follows that of PM_2.5_, suggesting that carbonaceous species are the major components of PM_2.5_. As shown in Table 1, OC concentrations at Lahore varied from 11.9 to 110.1 µg/m^3^ in wintertime and 4.5–29.7 µg/m^3^ in summertime, with mean values of 50.7 µg/m^3^ and 14.6 µg/m^3^, respectively. The OC/EC ratio is an effective method for determining the sources of carbonaceous aerosol [59]. The average OC/EC ratio during the winter and summer was 2.1 and 1.9, respectively. The OC/EC ratio significantly differentiates coal combustion (0.3–7.6), biomass burning (4.1–14.5), and automobile exhaust (0.7–2.4) [60,61]. The OC/EC ratio for diesel/gasoline-powered vehicle emissions is 1.0–4.2, whereas for wood combustion it is 16.8–40.0 [62,63].

The higher OC/EC ratio and significant correlation between OC and EC suggested that both OC and EC were emitted from the same sources, such as combustion of fossil fuels and biomass burning. The high OC/EC ratio during the wintertime has several reasons. These include biomass combustion for heating purposes in winter, condensation of semi-volatile organics into particles, and the formation of secondary organic carbon (SOC) under stagnant meteorological conditions [64]. The low temperature and stable atmosphere in winter enhance the condensation of volatile organic compounds on PM [65]. Comparatively, the lower OC/EC ratio and weak OC and EC correlation in summertime indicated that OC and EC emitted from different sources in summertime. The influence of weather conditions along with less coal burnt for heating and cooking purposes, as well as fossil fuel combustion used in transportation and small-scale industries, such as brick kilns, are also the possible reasons for the lower OC/EC ratio. OC is emitted directly as primary organic carbon (POC) and can also be formed as secondary organic carbon (SOC) by photochemical processes in the atmosphere. In both the winter and summertime, the OC/EC ratio was approximately 2.1 and 1.9. SOC was therefore measured using the EC tracer method [SOC = OC − EC × (OC/EC)_min_] [66]. The average concentration of SOC in winter and summertime sampling was 15.7 µg/m^3^ and 6.0 µg/m^3^, respectively. The high concentration of SOC in the wintertime can be attributed to the combustion of fossil fuels and biomass, which increases the emission of SOC precursor gases. Several studies have reported that an OC/EC ratio higher than 2.0 indicates the formation of SOA [67].

### 3.3. Metal Elements

Up to 95% of the measured elements consisted of metals, such as Al, Pb, Zn, Fe, and Mg. Li, V, Cr, Mn, Cu, Cd, Ti, As, Bi, Sn, Sr, Ba, and Ni are the metals with lower concentrations reported in the current study. In the current study, the mean concentration of the major elements, such as Al, Pb, Zn, Fe, and Mg, was significantly higher than that reported in previous studies in Lahore [68], Karachi and Kashmore, Sindh [69], and Peshawar [52]. Among the major metal elements, aluminum (Al) is mainly derived from resuspended road dust and wind-blown soil dust. Rubber industries, plastics, alloys, and battery units emitted lead (Pb) [70]; these sources are common on both small and large scales in the studied city. Additionally, the high concentration of Pb can also be emitted by the combustion of fuel in vehicles, which is toxic to animals and humans [71]. The high concentration of zin (Zn) is due to tire wear and fuel combustion in cars, buses, trucks, rickshaws, motorcycles, etc. [72]. At Lahore, the combustion of lubricating oil is the major source of zinc emission from motorcycles and rickshaws [73]. Iron (Fe) is a good indicator of the steel industry or metallurgy, and steel mills are located in Lahore [74].

### 3.4. Hydroxyl Radical (^•^OH) Generation (Oxidative Potential)

In Figure 2, the volume normalized and mass normalized OH radical generation in PM_2.5_ during the summer and wintertime sampling at Lahore is shown. The OH radical generation exhibits a similar daily variation. In winter and summertime, the average volume-normalized OH radical generation was 52.9 ± 25.7 and 33.9 ± 8.3 pmol/min/m^3^, respectively. The generation of ^•^OH increases as the concentration of PM_2.5_ increases, but the trend becomes flat when the concentration of PM_2.5_ reaches a higher level. This is in response to the extensive dose–response in epidemiological research showing the relationship between PM_2.5_ exposure and health risks, including heart disease, hypertension, and cardiovascular disease [75]. This indicates that less reactive oxygen species (ROS) are generated for every 10 µg/m^3^ increase in PM_2.5_ concentration [76]. A similar trend has also been observed in Seoul (Korea) [14], Beijing, and Wangdu (China) [76]. There are two possible reasons for this trend. Firstly, there is little influence on OH radical generation by an increase in PM_2.5_ mass. Secondly, the OH radical generation may be hindered because the reaction to produce OH radicals is reversible due to the high amount of redox-active metals in the extraction solution. In summertime, secondary inorganic ions (sulfate, nitrate, and ammonium) contributed more to the identified mass of PM_2.5_. However, these inorganic ions are not the main species contributing to ROS generation due to their non-redox activity [77]. Some studies reported that instead of the non-redox active components of PM_2.5_, the redox active components induced the OH radical generation [78]. Therefore, OH radical generation in summertime sampling is lower than in wintertime.

In the summertime, there was a strong correlation between OH radicals and WSOC, Cr, Ni, Zn, As, Cd, and Pb. In contrast, Mn, Fe, OC, and SOC strongly correlate with OH radicals in the wintertime. There is a strong correlation between the OH radicals and the components of PM_2.5_ associated with combustion sources. It is assumed that combustion sources, such as vehicle emissions and coal combustion, significantly contribute to OH radical generation. The correlation between OH radical and metals digested with strong acid shows that coal combustion and industrial emissions from nearby industrial zones are the two main sources for the OH radical generation [79]. Hydrogen peroxide (H_2_O_2_) enhances the transfer of electrons to form OH radicals. EPFRs transfer electrons to dissolved oxygen to form O_2_^−•^ and then H_2_O_2_ and finally generate OH radicals by transferring electrons from EPFRs to H_2_O_2_ [56]. The OH radicals can also be generated without H_2_O_2_ because of the contribution of EPFRs. The transition metals can be the main contributors to OH radicals’ formation because they catalyze the formation of OH radical via a Fenton-like reaction [13]. Lung diseases, such as asthma, chronic obstructive pulmonary disease (COPD), and respiratory distress syndrome, are due to oxidative stress; and hydrogen peroxide is a marker of oxidative stress [80]. So, it is clear from the correlation analysis of OH radicals and PM_2.5_ components in Section 4 that the oxidative potential (OP) of PM_2.5_ in both summer and wintertime is caused by combustion sources. More studies will be conducted to further evaluate the OP of PM_2.5_ in other parts of the country.

## 4. Correlation Analysis

### 4.1. EPFRs Correlation with Carbonaceous Species

A correlation analysis was carried out between the atmospheric EPFR concentrations and the carbonaceous fractions of PM_2.5_, such as EC, OC, WSOC, OC1, OC2, OC3, OC4, EC1, and EC2, to understand the sources and formation mechanisms of EPFRs. In Table 1, the results of the correlation analysis and the average concentrations of OC, EC, WSOC, SOC, and carbon fractions (OC1, OC2, OC3, OC4, EC1, EC2, and EC3) of PM_2.5_ are shown. The sources of each carbon fraction are specific and can thus be used as source indicators [81]. The EPFRs exhibited a significant correlation with EC and OC1 and a moderate correlation with OC, OC3, and EC3 during the winter season in Lahore. The sources of OC include fossil fuel and biomass combustion, whereas EC1 and OC3 are reported components of vehicle emissions and restaurant fumes, respectively [82]. Similarly, OC and EC3 are the markers of SOC formation and regional transport [83]. Higher OC3 concentrations indicated the contribution of road dust. OC3 was also found in cooking-related emissions, suggesting that the concentration of OC3 in the study area can also be influenced by the number of snack shops in the surrounding area [84]. The correlation analysis indicates that combustion sources are the major sources of EPFRs in PM_2.5_ during the winter.

The summertime EPFRs were significantly correlated with SOC, OC1, and OC2. Combustion processes at higher temperatures, such as coal combustion and vehicle emissions, contribute to OC2 and EC3, also known as soot-EC. In addition, OC2 and EC3 can also be formed through gas-to-particle conversion [81]. OC2 is a good coal combustion tracer and is abundant in coal-fired power plant emissions [85]. The correlation analysis indicated that vehicle emissions and coal combustion are the major sources of EPFRs during the summertime. The correlation analysis of EPFRs with carbonaceous components of PM_2.5_ suggests that coal combustion, biomass burning, and vehicle emissions are the major sources of EPFRs in the winter and summertime.

### 4.2. Correlation between ^•^OHv and PM_2.5_ Components

The chemical species of PM_2.5_ responsible for the OH radical generation can be identified using correlation analysis. In the wintertime sampling, OC and SOC show a correlation with volume normalized OH radical (^•^OHv), commonly associated with traffic pollution, whereas WSOC show correlation with OH radical in the summer [86]. EC and ^•^OHv have no correlation in winter and summertime sampling. This could be due to the different compositions and sources of PM_2.5_, particularly carbonaceous species in both winter and summertime sampling. Moreover, the EC and ^•^OHv weak correlation in winter is due to the correlation of OC and ^•^OHv, because OC and EC are usually emitted from the same sources, such as vehicle emissions [87]. Some studies reported that the oxidative potential of freshly emitted soot (EC, BC) is low, but it increases when O_3_ oxygenated the soot particles or SOC and PAH adsorbed on the surface of soot particles [45].

The correlation of volume normalized OH radical with the metal elements and carbonaceous species is shown in Table 3. During the summer, Cr, Ni, As, Cd, and Pb have different correlation strengths with OHv radicals, whereas Mn and Fe have strong correlations with OHv radicals during the winter. The coal combustion tracers are OC, EC, As, and Mn [88]. Zn, Fe, OC, and EC are emitted by vehicle emissions and brake wear [89]. OHv radicals correlated significantly with Fe, Mn, Cr, Ni, Cd, Pb, OC, SOC, and WSOC in both summer and winter. This indicates that combustion sources, such as vehicle emissions and coal combustion, contributed to OH radical generation [82]. Vehicle emissions could be a significant factor contributing to the oxidative potential (OP) of PM_2.5_ in the urban areas of Lahore. The redox-active metals, such as Fe and Mn, can be derived from dust and other sources. It indicates that road dust and soil in both seasons can also contribute to the OP of PM_2.5_ in Lahore [90]. In both seasons, a significant correlation between volume normalized OH radicals and metals indicated that coal combustion and industrial emissions from nearby industrial zones contribute to the OHv radical generation. Therefore, the correlation analysis showed that combustion sources, such as vehicle and industrial emissions, significantly contributed to the OP of PM_2.5_ in both summer and winter.

## 5. Health Effects

### Atmospheric EPFRs Exposure Risk

PM_2.5_ enters the human body primarily through inhalation. The solution in the pulmonary alveoli is the first to react with PM_2.5_ when it enters the lung. The risks of PM_2.5_-bound EPFRs in a healthy person can be assessed by an in vitro procedure that uses water, the main component of the lung physiological solution [91]. EPFRs in PM_2.5_ and cigarette tar have been shown in previous studies to cause similar toxicity in humans [35]. For instance, EPFRs can promote the generation of ROS, which can cause cardiovascular and respiratory disease [92] and damage DNA [35]. Gehling and Dellinger [35] used this method for the first time to calculate the equivalent cigarette number in 10 states of the United States. Thus, we converted EPFR emissions in PM_2.5_ into equivalent cigarette numbers to assess the exposure and health risks. As shown in Figure 3, the potential health effects of atmospheric EPFR exposure on the residents of Lahore were evaluated using the number of cigarettes smoked per person per day in the summer and wintertime sampling at Lahore. Lyons was the first to report that tobacco smoke also contained cancer-causing, long-lived free radicals. In both winter and summertime, the residents of Lahore were exposed to EPFRs through PM_2.5_ inhalation. The average EPFR inhaled per person per day was equivalent to 4.0 cigarettes in wintertime sampling and 0.6 in summertime sampling, which is equivalent to 1460.0 cigarettes per year in wintertime and 219.0 in summertime. In Xian, China, the amount of EPFR inhaled is equivalent to 4.7 cigarettes per person per day [44], slightly higher than the current study. Gahling and Dellinger [35] conducted a study in the most polluted county in the U.S., San Joaquin County. The EPFR exposure evaluation was equivalent to approximately 101 cigarettes smoked per person per year. This is significantly lower than the findings of the current study. The residents of Lahore are approximately 14.5 and 1.8 times more exposed to EPFRs than those in San Joaquin County, USA. In Wanzhou, Chongqing, China, the EPFR exposure risk reported was equivalent to 0.6–3.0 cigarettes smoked per person per day in summer, and 0.5–4.5 in winter [55]. In Pakistan, particularly in Lahore, extreme air pollution events occur during winter. The EPFR exposure risk in Lahore is significant, and more pathological studies on EPFRs need to be carried out. We recommend the findings of this study to epidemiological researchers to further determine the association between EPFR exposure and the intensity of cardiovascular and respiratory diseases.

## 6. Limitations of the Study

The current study has some limitations. In this study, a sampling campaign lasting only 15 days (day and night; 30 samples each campaign) was carried out in both wintertime and summertime. The small number of samples may have limited the accuracy of the data. For future research, long-term monitoring campaigns must be conducted in different megacities of Pakistan. Due to a lack of access to data, the concentration of PM_2.5_ in the current study was not compared to the online PM concentration in Pakistan. This study mainly focused on the emissions of EPFRs in PM_2.5_ and their exposure risk in the winter and summer in Lahore. Long-term campaigns are required to better understand the exposure risk of PM and EPFRs. Due to various anthropogenic and meteorological factors, the EPFRs and PM_2.5_ concentrations in Lahore may vary in the future. Therefore, future studies should refine EPFRs in PM_2.5_ from various combustion sources to develop more accurate EPFR emissions inventories for various seasons in urban sites in Lahore and other megacities.

## 7. Conclusions

High concentrations of PM_2.5_ in Lahore are currently a major air quality concern. In Lahore, the concentration of PM_2.5_ in both winter and summertime was significantly higher than the 24-h standard set by the US Environmental Protection Agency (US-EPA) and national environmental quality standards of Pakistan (Pak-NEQS). The EPFR concentration in wintertime sampling is higher than in summertime. Based on the average concentration of EPFR in the atmosphere, the amount of EPFR inhaled per person per day in Lahore during the winter and summertime sampling is equivalent to 4.0 and 0.6 cigarettes, respectively. Similarly, the residents of Lahore are 14.5 and 1.8 times more exposed to EPFR than San Joaquin County, United States. The g factor of EPFRs in PM_2.5_ was 2.0027–2.0032 in wintertime, and 2.0026–2.0033 in summertime. The g factor of EPFRs in both sampling periods similar to EPFRs from smoke and tar from coal, tobacco, and petroleum. This range of g factors represents carbon EPFRs that contain heteroatoms or free radicals. According to the correlation analysis between EPFRs and carbonaceous species, possible sources of EPFRs include coal combustion, vehicle emissions, industrial activities, etc.

This study also investigated the DTT-based OH radical generation in cell-free aqueous extracts of PM_2.5_ in the urban sites of Lahore during the winter and summertime sampling. PM_2.5_ generates more volume-normalized OH radicals than mass-normalized OH radicals. In Lahore, the wintertime PM samples generate more OH radicals than in the summertime. The components of PM_2.5_, particularly OC, EC, WSOC, Cr, Fe, Co, Al, Mn, and Ni, play a significant role in OH radical generation in both summer and wintertime sampling. Major sources of carbonaceous and elemental species of PM_2.5_ include biomass burning, vehicle emissions, and coal combustion. Thus, correlation analysis suggests that industrial and vehicle emissions in summer and wintertime play a significant role in OH radical generation. The sources of PM_2.5_ that contributed to its oxidative potential would help to investigate the health problems and the associated sources. The findings will also support the design of effective countermeasures and control strategies for air pollution.

The carbonaceous species are the dominant components of PM_2.5_ and the concentration of PM_2.5_ in both wintertime and summertime in Lahore was reported as 15 (wintertime) and 4.6 (summertime) times higher than NEQS of Pakistan and the WHO interim target-1 of PM_2.5_. The possible sources of EPFRs and OH radicals in both wintertime and summertime are the primary combustion sources, such as industrial activities, vehicle emissions, biomass combustion, and coal combustion. The concentration of EPFRs and OH radicals in Lahore was reported as significantly higher in wintertime than the summertime. Previous studies have shown that EPFRs in PM_2.5_ and in cigarette tar caused similar toxicity to humans [34]. Gehling and Dellinger [34] for the first time developed a method to convert EPFR concentration in PM_2.5_ into equivalent cigarette numbers to assess the exposure and health risk. Thus, in this study, the exposure risks of EPFRs in particles deposited in the human body were converted to the equivalent number of cigarettes inhaled per adult per day to evaluate their exposure and health risk. The average EPFR inhaled per person per day was equivalent to 4.0 cigarettes in wintertime sampling and 0.6 in summertime sampling, which is equivalent to 1460.0 cigarettes per year in wintertime and 219.0 in summertime.

## Figures and Tables

**Figure 1 ijerph-20-02384-f001:**
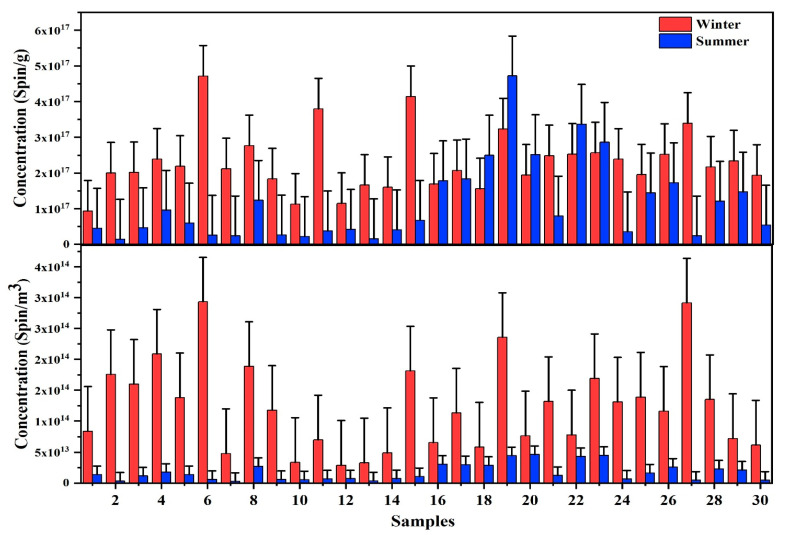
EPFRs concentration in PM_2.5_ mass and atmosphere in Lahore during winter and summertime sampling, 2019.

**Figure 2 ijerph-20-02384-f002:**
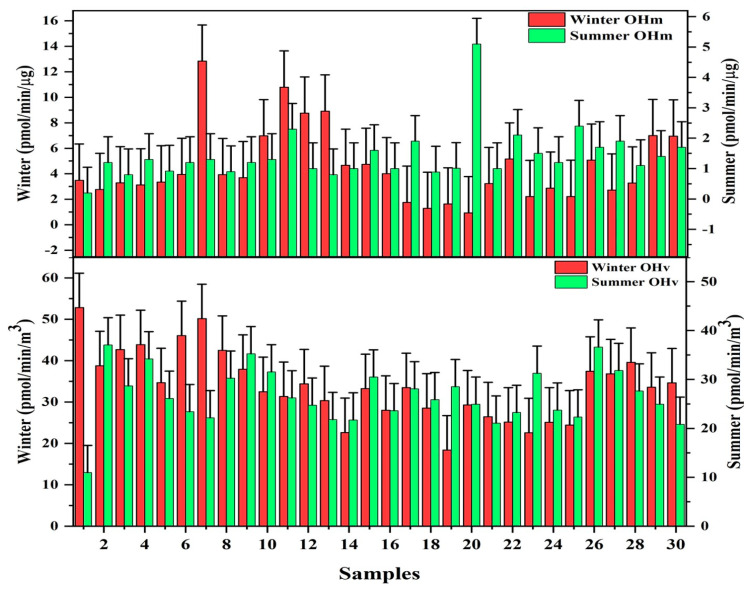
Hydroxyl radical (^•^OH) generation rate in the atmosphere and PM_2.5_ mass in Lahore during the sampling period of winter and summertime, 2019.

**Figure 3 ijerph-20-02384-f003:**
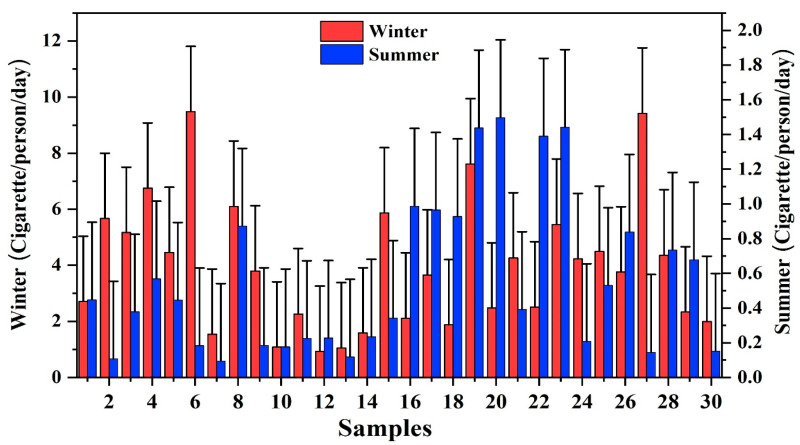
The amount of EPFR exposure equivalent to the number of cigarettes per person per day in Lahore in winter and summertime sampling, 2019.

**Table 1 ijerph-20-02384-t001:** Mean and range with a standard deviation of carbonaceous species (µg/m^3^) in PM_2.5_, EPFRs (spin/m^3^) and EPFRs (spin/g), and correlation coefficient (R) of carbonaceous species with EPFRs (spin/m^3^) in Lahore during winter and summertime sampling, 2019.

	Winter			Summer		
	Mean ± Stdv	Range	R	Mean ± Stdv	Range	R
TC	76.3 ± 46.9	16.3–171.2	0.82 **	23.2 ± 8.1	6.6–42.9	0.06
OC	50.7 ± 30.5	11.9–110.1	0.79 **	14.6 ± 5.6	4.5–29.7	0.34
EC	26.5 ± 18.0	4.5–64.9	0.80 **	8.6 ± 3.4	2.1–15.0	−0.23
WSOC	35.2 ± 14.9	14.7–69.9	0.75 **	7.4 ± 1.6	3.4–11.1	−0.32
SOC	15.7 ± 11.3	0.1–43.7	0.45 *	6.0 ± 4.5	0.3–22.4	0.65 **
OC1	30.6 ± 24.0	4.0–89.0	0.84 **	0.4 ± 0.6	0.0–3.2	0.76 **
OC2	158.1 ± 39.9	59.6–101.5	−0.67 **	5.3 ± 2.2	2.0–12.5	0.57 **
OC3	39.1 ± 26.6	5.6–90.8	0.74 **	5.3 ± 2.0	2.2–10.5	0.14
OC4	117.5 ± 29.5	79.2–194.4	−0.50 **	3.6 ± 2.1	0.2–9.6	0.02
EC1	43.4 ± 52.1	3.5–201.0	0.52 **	0.9 ± 0.7	0.0–3.1	−0.06
EC2	166.3 ± 159.7	1.3–499.0	−0.13	5.4 ± 2.6	0.3–11.9	−0.26
EC3	14.4 ± 30.8	0.0–132.5	0.54 **	2.2 ± 1.2	0.1–4.5	−0.09
EPFRs (spin/m^3^)	1.2 × 10^14^ ± 7.2 × 10^13^	2.9 × 10^13^–2.9 × 10^14^		1.7 × 10^13^ ± 1.4 × 10^13^	2.9 × 10^12^–4.6 × 10^13^	
EPFRs (spin/g)	2.3 × 10^17^ ± 8.6 × 10^16^	9.3 × 10^16^–4.7 × 10^17^		1.1 × 10^17^ ± 1.1 × 10^17^	1.4 × 10^16^–4.7 × 10^17^	
g-value	2.0030	2.0027–2.0032		2.0029	2.0026–2.0033	

Total carbon (TC) = OC + EC; Level of significance: *: *p* < 0.05; **: *p* < 0.01

**Table 2 ijerph-20-02384-t002:** Comparison of PM_2.5_ mass concentration (μg/m^3^) of the present study with other studies conducted in Megacities of Pakistan and India.

City	Season	PM_2.5_	Reference
Lahore	Summer	170 µg/m^3^	[33]
Lahore	Winter	191 µg/m^3^	[48]
Varanasi	Winter	229.7 µg/m^3^	[49]
Faisalabad	Winter	209 µg/m^3^	[50]
Amritsar	Winter	147.6 µg/m^3^	[51]
Delhi	Winter	357.3 µg/m^3^	[51]
Peshawar	Winter	286 µg/m^3^	[52]
Peshawar	Winter	172 µg/m^3^	[53]
Lahore	Winter	522.2 µg/m^3^	This Study
Lahore	Summer	162.5 µg/m^3^	This Study

**Table 3 ijerph-20-02384-t003:** Correlation coefficient (R) of volume-normalized OH radical (OH_v_) with chemical components of PM_2.5_ in Lahore during winter and summertime sampling, 2019.

	Winter	Summer
OC	0.40 *	0.27
EC	0.22	−0.20
WSOC	0.33	0.45 *
SOC	0.62 **	0.15
Cr	−0.12	0.51 **
Mn	0.55 **	0.32
Fe	0.54 **	0.31
Ni	−0.12	0.49 **
Cu	0.34	−0.20
Zn	0.34	0.41 *
As	0.35	0.59 **
Cd	−0.12	0.48 **
Pb	0.16	0.49 **

Level of significance: *: *p* < 0.05; **: *p* < 0.01

## Data Availability

The original contributions are presented in the study; further inquiries can be directed to the corresponding authors. Data used in the present study can be obtained by contacting sirima.p@chula.ac.th.

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
