# Peer review of "Characteristics and Risk Assessment of Environmentally Persistent Free Radicals (EPFRs) of PM_2.5_ in Lahore, Pakistan"

_ijerph, 2023, doi:10.3390/ijerph20032384_

Round 1
Reviewer 1 Report
The article is within the scope of the journal, it did a lot of research, but it needs many improvements so that it can be accepted.
Comments:
Ciatation must be on square brackets and not on (Cohen et al., 2017) form.
you need to cite more current references (70% of the last 5 years).
Are ROS the free radical forms of oxygen? This terminology is not common, prefer to use the technical term O..
“Environmentally persistent free radicals (EPFRs)”, Give preference to quoting some of these free radicals, rather than using unusual terminology. It is more common to talk about primary, secondary and tertiary pollutants, the latter being the result of chemical reactions (which occur with these free radicals and/or under the effect of ionizing radiation).
Methods
You must cite at least one reference for each subitem of the methodology
Results
Your results are very unusual. It is usual to evaluate the impacts of PM, CO, SO2, O3, NOx, HC and VOC, as indicated by the WHO. There are even tables with the possible diseases caused by exposure to one or more of these components, as you can see in this article:
José Carlos Curvelo Santana 1,2,*, Amanda Carvalho Miranda 1 , Luane Souza 2 , Charles Lincoln Kenji Yamamura 1 , Diego de Freitas Coelho 3 , Elias Basile Tambourgi 3 , Fernando Tobal Berssaneti 1 and Linda Lee Ho. Clean Production of Biofuel from Waste Cooking Oil to Reduce Emissions, Fuel Cost, and Respiratory Disease Hospitalizations. Sustainability 2021, 13, 9185. https://doi.org/10.3390/su13169185
José Carlos Curvelo Santana 1, *, Amanda Carvalho Miranda 2, Charles Lincoln Kenji Yamamura 1, Silvério Catureba da Silva Filho 2,3, Elias Basile Tambourgi 3, Linda Lee Ho 1 and Fernando Tobal Berssaneti. Effects of Air Pollution on Human Health and Costs: Current Situation in São Paulo, Brazil. Sustainability 2020, 12, 4875; doi:10.3390/su12124875
Therefore, I suggest that in your discussion you emphasize that the PM, HC and VOC that make up the OC and EC can cause these diseases (see table). And that the oxidation of HC and VOC by radiation and the reaction with O3 can increase the concentration of PM2.5 which are the most dangerous to human health.
Mention in the methodology the emission limits that your country uses and make a discussion in relation to WHO.
5.1. Your references are very old and consequently, statements like these are not valid: “Lyons et al. (1958) reported 400 that tobacco smoke also contained cancer-causing long-lived free radicals.” Do not associate environmental pollution with cigarette smoke. Because the contribution is exclusively from vehicle emissions, energy generation and industries. For this reason, I think these results are not true and compromise your research. I'm sorry for the sincerity!
I strongly suggest eliminating item 5, as it threatens to reject your article. Use the discussion I suggested regarding the most common pollutants.
Conclusions
Its conclusions are leaner and more objective than its discussions.
References
The form of presentation of the references is not correct. View the guide for authors
Author Response
We would like to express our sincere gratitude for offering us the opportunity to revise and resubmit our manuscript. We also would like to let you know that all the issues raised by appointed reviewers have been fully addressed, and the manuscript has been revised accordingly. We have included a point-by-point response to the reviewers’ comments and highlighted the changes made in the manuscript.
Responses to the reviewers’ comments:
Reviewer 1:
“1. Citations must be on square brackets and not on (Cohen et al., 2017) form.”
Response: Thanks for the reviewer's suggestion. The citations within the text has been changed to square bracket form throughout the manuscript.
“2. you need to cite more current references (70% of the last 5 years).”
Response: Thanks for the reviewer’s comment. We have changed the citations and added the current references.
“3. Are ROS the free radical forms of oxygen? This terminology is not common, prefer to use the technical term O.”
Response: Thanks for the reviewer’s comment. ROS are oxygen-containing, reactive molecules such as hydroxyl radical, superoxide and peroxides. With an increase in ROS, the ability of a biological system to detoxify the reactive species is eventually overridden which triggers oxidative stress and causes tissue damage. Biological research has shown that EPFRs have generated ROS and are leading to oxidative stress causing heart and lung damage. The generation of ROS induces imbalance between antioxidants and oxidants and thus causes several cardiopulmonary diseases (Amatullah et al., 2012). The ability of aerosol to generate ROS is also called its oxidative potential (OP) (Hopke, 2003).
“4. “Environmentally persistent free radicals (EPFRs)”, Give preference to quoting some of these free radicals, rather than using unusual terminology. It is more common to talk about primary, secondary and tertiary pollutants, the latter being the result of chemical reactions (which occur with these free radicals and/or under the effect of ionizing radiation).”
Response: Thanks for the reviewer’s comment. Free radicals are atoms or groups containing unpaired electrons, they usually have strong chemical reactivity and short lifetimes. Free radicals with long lifetimes (months or even years) in the environment are currently called environmentally persistent free radicals (EPFRs), which have received much attention in recent years as new environmentally hazardous substances (Vejerano et al., 2018; Chen et al., 2019c). EPFRs are present in different environmental media, such as water and soil, and even in the atmosphere (Dellinger et al., 2001; Truong et al., 2010; Vejerano et al., 2012).
“5. You must cite at least one reference for each subitem of the methodology.”
Response: Thanks for the reviewer's suggestion. The citations have been added to each sub section of methodology. For detail please refer to the page 3 and 4.
“6. Your results are very unusual. It is usual to evaluate the impacts of PM, CO, SO2, O3, NOx, HC and VOC, as indicated by the WHO. There are even tables with the possible diseases caused by exposure to one or more of these components, as you can see in this article.”
Response: Thanks for the reviewer’s comment. This study has been carried out on PM2.5 and environmentally persistent free radicals (EPFRs) and its exposure risk. The studies on this issue has already been carried out in several regions like China, USA, Germany, etc. The results of the current study are consistent with previous studies carried out in China and USA.
“7. Mention in the methodology the emission limits that your country uses and make a discussion in relation to WHO.”
Response: Thanks for the reviewer’s comment. The national environmental quality standards (NEQS) of Pakistan and WHO for PM2.5 has already been discussed and cited in results and discussion section. Please refer to page 4, line 182.
“8. Your references are very old and consequently, statements like these are not valid: “Lyons et al. (1958) reported 400 that tobacco smoke also contained cancer-causing long-lived free radicals.” Do not associate environmental pollution with cigarette smoke. Because the contribution is exclusively from vehicle emissions, energy generation and industries. For this reason, I think these results are not true and compromise your research. I'm sorry for the sincerity!
I strongly suggest eliminating item 5, as it threatens to reject your article. Use the discussion I suggested regarding the most common pollutants.”
Response: Thanks for the reviewer’s comment. Lyons et al (1960) for the first time found EPFRs in cigarette tar and is considered one of the health risk factors in cigarette smoke. EPFRs in PM2.5 and cigarette tar have been shown in previous studies to cause similar toxicity in humans. EPFRs can promote the generation of ROS, which can cause cardiovascular and respiratory disease and damaging DNA (Gehling & Dellinger, 2013). Gehling and Dellinger, (2013) for the first time developed a method to convert EPFR concentration in PM2.5 into equivalent cigarette numbers to assess the exposure and health risk. Thus, in this study, the exposure risks of EPFRs in particles deposited in the human body were converted to the equivalent number of cigarettes inhaled per adult per day to evaluate their exposure and health risk.
Reviewer 2 Report
General comment.
The paper entitled “Characteristics and risk assessment of environmentally persistent free radicals 2 (EPFRs) of PM2.5 in Lahore, Pakistan” is in the scope of the journal and may be recommended for publishing after major revision.
Specific comment:
The author uses an attractive title that helps to convey the main topic of the study. However, the abstract should be reconsidered. It needs major revision mainly about findings and perspective to provide information encouraging readers to read the full paper.
The keywords in the abstract section should be reconsidered and more explicit. For instance, remove the keyword “Concentration”.
Authors can add a section in the introduction showing on it the main questions that this work need to answer. Then at the final of the paper show how can find an answer to that question based on their results and experiment. This is a very important point to make in this paper as a guide for readers.
L70-84: The authors stated only studies conducted in Europe, North America, and Asia. However, they should add also those carried out in African locations such as Tahri et al., 2017 (Seasonal variation and risk assessment of PM2.5 and PM2.5–10 in the ambient air of Kenitra, Morocco), Benchrif et al 2022 ( Aerosols in Northern Morocco-2: Chemical Characterization and PMF Source Apportionment of Ambient PM2.5), Lemou et al. 2020 (Chemical characterization of fine particles (PM2.5) at a coastal site in the South Western Mediterranean during the ChArMex experiment)….
Section 2.1:
Please add the air volume sampler type, filter diameter, and sampling period.
The sampling period was stated as " 11:30 h ". The authors should explain and provide more details about sampling frequency, sampling periods, ….
Section 2.2: Add proper citation of the electron paramagnetic resonance (EPR)
Section 3.1:
The authors stated that the meteorological conditions in the wintertime at Lahore are poor. More descriptions are required.
More details should be given to explain g-factor. And discuss deeply the variation in the g-factor with a concentration in different seasons.
L184: when more than 2 references are used please shortly describe each in the text, otherwise remove unnecessary ones.
Table 1: What does ** stand for? please change R2 to R. I don’t understand why * is used for the correlation between SOC and EPFR.?
Section 3.2: the authors should also discuss the OC/EC ratios in the summertime.
Section 3.3: It would be more appropriate to investigate the Enrichment factor. The source identification approach based on metal concentrations is poor.
I suggest adding a new section/paragraph including the hypothesis and limitations used in this study.
Figure 2 quality should be improved.
Table 3: Does R refer to the spearman correlation coefficient?
Section 4.1: The authors concluded that the source characteristics of EPFRs are affected by seasonal characteristics. what can be the implication of such seasonality? what is the driven factor?
Author Response
We would like to express our sincere gratitude for offering us the opportunity to revise and resubmit our manuscript. We also would like to let you know that all the issues raised by appointed reviewers have been fully addressed, and the manuscript has been revised accordingly. We have included a point-by-point response to the reviewers’ comments and highlighted the changes made in the manuscript.
Responses to the reviewers’ comments:
Reviewer 2:
“1. The author uses an attractive title that helps to convey the main topic of the study. However, the abstract should be reconsidered. It needs major revision mainly about findings and perspective to provide information encouraging readers to read the full paper.”
Response: Thanks for the reviewer's comment. We have rewritten the abstract, add more details of the findings of the study to make it more informative. For reference, see the track change part on page 1.
“2. The keywords in the abstract section should be reconsidered and more explicit. For instance, remove the keyword “Concentration”.”
Response: Thanks for the reviewer’s suggestion. We have changed the keywords. Please refer to page 1.
“3. Authors can add a section in the introduction showing on it the main questions that this work need to answer. Then at the final of the paper show how can find an answer to that question based on their results and experiment. This is a very important point to make in this paper as a guide for readers.”
Response: Thanks for the reviewer's comment. We added the research questions in the last paragraph of introduction. Please see the research questions on page 2 of the manuscript.
“4. L70-84: The authors stated only studies conducted in Europe, North America, and Asia. However, they should add also those carried out in African locations such as Tahri et al., 2017 (Seasonal variation and risk assessment of PM2.5 and PM2.5–10 in the ambient air of Kenitra, Morocco), Benchrif et al 2022 ( Aerosols in Northern Morocco-2: Chemical Characterization and PMF Source Apportionment of Ambient PM2.5), Lemou et al. 2020 (Chemical characterization of fine particles (PM2.5) at a coastal site in the South Western Mediterranean during the ChArMex experiment)…”
Response: Thanks for the reviewer's suggestion. We have added the studies from Africa in the results and discussion section to compare with present study. Please see the above-mentioned cited studies on page 4 of the manuscript.
“5. Please add the air volume sampler type, filter diameter, and sampling period.”
Response: Thanks for the reviewer's comment. The sampler type, filter diameter, and sampling period has been added to the methodology section. Please see the changes on page 3.
“6. The sampling period was stated as " 11:30 h ". The authors should explain and provide more details about sampling frequency, sampling periods, ….”
Response: We deeply appreciated the reviewer’s comment. We have added the details of the sampling periods. The sampling was carried out day and night time, therefore, the sampling frequency is shown 11:30. The details added to the methodology sections of the manuscript. Please refer to page 3.
“7. Add proper citation of the electron paramagnetic resonance (EPR).”
Response: Thanks for the reviewer’s suggestion. We have added the proper citation of the electron paramagnetic resonance (EPR). Please see the change on page 3.
“8. The authors stated that the meteorological conditions in the wintertime at Lahore are poor. More descriptions are required.”
Response: Thanks for the reviewer’s comment. We have added more description of meteorological conditions in the wintertime at Lahore on page 3 of the manuscript.
“9. More details should be given to explain g-factor. And discuss deeply the variation in the g-factor with a concentration in different seasons.”
Response: We appreciate the reviewers’ comment. The details about g-factor has been added in the manuscript. Please see page 6 for the manuscript.
“10. when more than 2 references are used please shortly describe each in the text, otherwise remove unnecessary ones.”
Response: Thanks for the reviewer's suggestion. The unnecessary reference has been removed.
“11. Table 1: What does ** stand for? please change R2 to R. I don’t understand why * is used for the correlation between SOC and EPFR?”
Response: We appreciate the reviewer for this valuable comment. * and ** represent the level of significance such as *: p <0.05; and **: p <0.01. we add it to the table as a footnote. We changed R2 with R. The measured OC not only come from the direct emissions of particles as primary organic carbon (POC) but also in the form of SOC formed by the chemical processes in the atmosphere (Shen et al., 2017). In addition to primary sources such as biomass and coal combustion, secondary chemical processes in the atmosphere may also be an important source of EPFRs in atmospheric PM (Chen et al. 2019b and 70 2019d; Tong et al., 2018).
“12. The authors should also discuss the OC/EC ratios in the summertime.”
Response: Thanks for the reviewer's comment. We have added the discussion about OC/EC in the summertime on page 7 of the manuscript.
“13. It would be more appropriate to investigate the Enrichment factor. The source identification approach based on metal concentrations is poor.”
Response: Thanks for the reviewer's suggestion. This study mainly focused on EPFRs and its exposure risk, for this reason we use the source identification approach based on metal concentrations, which has already been identified in previously conducted studies.
“14. I suggest adding a new section/paragraph including the hypothesis and limitations used in this study.”
Response: Thanks for the reviewer's suggestion. We have added a new section named limitations of the study on page 12.
“15. Figure 2 quality should be improved. Does R refer to the spearman correlation coefficient?”
Response: Thanks for the reviewer's suggestion. We have changed the figure 2 with high quality image. The R refer to the Pearson’s correlation coefficient.
“16. The authors concluded that the source characteristics of EPFRs are affected by seasonal characteristics. what can be the implication of such seasonality? what is the driven factor?”
Response: Thanks for the reviewer’s valuable comment. The sources of PM2.5 in winter and summertime are quite different. In wintertime, besides, the vehicular and industrial emissions, fossil fuel, wood, and coal etc. used for the heating and cooking purposes. In wintertime, the meteorological conditions are stagnant in Lahore with low wind speed, temperature, and lower atmospheric boundary layer. While in summertime the major sources are considered industrial and vehicular along with long range transport of pollutants from the western states of India. In summertime, there is high wind speed, temperature, and higher boundary layer. The driving factors, including traffic congestion, domestic heating, population intensity, topography, and meteorology.
Round 2
Reviewer 1 Report
I accept the manuscript on present form
Reviewer 2 Report
The research work shown in this article is interesting and well-founded. The paper has been revised, although, there is still room for improvement, especially in the discussion of what is the contribution of long-range transport on PM2.5 concentration and then on EPFRs. Overall, the paper has been rigorously revised. As such, the reviewer recommend that this paper be considered for this journal.